# Region matters: Mapping the contours of undernourishment among children in Odisha, India

**Apoorva Nambiar[1], Satish B. Agnihotri [2]\*, Ashish Singh[3], Dharmalingam Arunachalam[4]**

**1** IITB-Monash Research Academy, Indian Institute of Technology Bombay, Powai, Mumbai, India, **2** Centre for Technology Alternatives for Rural Areas, Indian Institute of Technology Bombay, Powai, Mumbai, India, **3** Shailesh J. Mehta School of Management, Indian Institute of Technology Bombay, Powai, Mumbai, India, **4** School of Social Sciences, Monash University, Melbourne, Victoria, Australia

\* sbagnihotri@iitb.ac.in

## Abstract

### Background

Levels of child undernutrition and its correlates exhibit considerable spatial variation at different levels of granularity. In India, such variations and their interrelation have not been studied at the sub-district level primarily due to the non-availability of good quality granular data. Given the sheer regional diversity in India, it is essential to develop a region-specific evidence base at the micro-level.

### Data and objectives

The current study utilised, for the first time, a sub-district level survey data (Concurrent Child Monitoring Survey-II, 2014–15) to investigate the statistically significant clusters and spatial patterns of burden of undernutrition among children. The emergence of distinct patterns at the level of natural geographical regions of the state–coastal, southern and northern regions, lead to a region-specific analysis to measure the impact of various demographic, socio-economic and maternal factors on the prevalence of undernutrition specific to the three regions, using the National Family Health Survey-IV unit-level data.

### Methods

The spatial dependence and clustering of child undernourishment across sub-districts in Odisha were studied using various spatial statistical techniques, including spatial econometric models. Binary logistic regression was applied in the region-specific analysis.

### Results

Findings indicated statistically significant spatial clustering of undernutrition among children in specific geographic pockets with poor sanitation, low institutional and skilled deliveries, poor maternal health reinforcing the need for inter-sectoral coordination. Disparities across the three natural-regions, suggest that the parameters requiring priority for intervention may differ across levels of overall development.

**Data Availability Statement:** (1) The National Family Health Survey Data used is publicly available on the Demographic and Health Survey (DHS) website: https://dhsprogram.com/data/ dataset/

India_Standard-DHS_2015.cfm?flag=1. (2) The
Odisha Concurrent Child Monitoring Survey Data
was used in the current study with approval in
principle by WCD & MS Department, Govt. of
Odisha. The report of the survey can be accessed
at http://www.nrhmorissa.gov.in/writereaddata/
Upload/Documents/Concurrent%20Monitoring-%
20Round-IIFinal.pdf.

**Funding:** The authors received no specific funding
for this work.

**Competing interests:** The authors have declared
that no competing interests exist.

## Conclusion

The spatial clustering of different socio-demographic indicators in specific geographic pockets highlights the differential impact of these determinants on child undernutrition thereby reinforcing a strong need for targeted intervention in these areas. Present analysis and the evidence-based micro-level analysis can be utilised as a model for other Indian states and low-resource countries, making interventions more effective through multiple, synergistic and a multi-sectoral approach.

## Introduction

The persistence of high level of child undernutrition in India, and the slow rate of reduction, has been a source of worry at both policy and academic levels. Notwithstanding its Integrated Child Development Scheme (ICDS), the most extensive intervention of its kind globally, running for over four decades now, the country has missed its child undernutrition targets under the Millennium Development Goals [1]. Estimates from the National Family Health Survey (NFHS) (3rd, 4th and 5th rounds) have confirmed the reduction, but at a slow rate, in the rates of malnutrition indicators [2–4]. The current asking rate of reduction to meet the Sustainable Development Goals (SDG) targets of achieving zero hunger and malnutrition by 2030 [5] and those set under the National Nutrition Mission—reduction in stunting, underweight, anaemia and low birth weight by 2%, 2%, 3% and 2% per annum, respectively [6], also appears daunting.

Availability of good quality data at an adequately granular level has always been a problem. The routinely collected data from the field under the ICDS is fraught with quality issues while data from the NFHS used to be available at a 10-year interval and that too at the level of a state or a union territory. The 4th round of NFHS, for the first time, provided data on child nutritional status and its correlates at district-level and the recent 5th round of survey too, which has followed a quicker interval of four years. This provided a good opportunity for researchers to study district-level variations across India using many aspects of geo-coding and small area estimation techniques and facilitated more in-depth analyses [7, 8]. Thus, the availability of NFHS-4 & 5 district-level data has led to a strong appreciation of regional diversity in child undernutrition patterns as well as its correlates. Studies have also helped bring out the significance of convergent action and preventive steps to tackle the issue of child malnutrition [9].

Though these have resulted in a shift of emphasis from state-level planning and intervention to district-level planning and convergent action between different sectors in the last few years, as reflected in the National Nutrition Mission strategy [6], this is not enough to understand the ground-level reality. The findings at a sub-district level have still eluded the policy and scholarship. This is important since spatial variation in child undernutrition and its correlates can be considerable even within a district. Besides this, the ICDS projects are primarily coterminous with the sub-district administrative levels called the blocks. When individual data is collected at the level of local governance units and aggregated upwards to the national level, the level at which decision-making mostly happens, the granularity of data disappears in the process. Micro-level disparities are masked by state-level or district-level averages. Collecting data at the micro-level and further analyses are therefore critical to reaching the nutritional targets set for India. It will help the local and regional decision-makers (e.g, Block Development Officers, Members of Panchayats, etc.) to understand the ample disparities at granular

levels, targeting local development issues in each of these areas and understand the successes and failures of policies and interventions at ground level. Availability of such data would be useful to analyse the patterns and correlates of child undernutrition at a granular level, highlight the spatial disparities and to identify those enclaves which needs prioritisation, which the current paper has focused.

Additionally, geospatial analysis has been adopted to understand the variations in undernutrition at different levels- regional, state, district, urban-rural distinctiveness and high-low density in Indian studies [10–12]. However, these studies point out the lack of nationally representative individual-level data. NFHS-4 removed this limitation to a considerable extent. NFHS-4 data was used to look at the geographic burden and spatial differences in the distribution of childhood stunting and undernutrition at the district level [9, 13]. Using the same data, Parliamentary Constituency (PC) wise estimates were generated using GPS dataset of the sampling clusters, and by using boundary shapefiles, geographical crosswalks between districts and PCs were constructed [14]. This methodology was again advanced and adapted using the precision-weighted estimation method and hierarchical logistic regression modelling techniques to develop complex survey design and sampling variability to study child malnutrition patterns [15]. Also, LISA and Moran's statistics were used in studies to point out different levels of association of the malnutrition indicators among children under five years and other correlates which have established different spatial patterns pertaining to children malnourished against the mesoscale correlates in the geographical hotspots of India [16], but even this study was at district level only. The method of precision mapping was used to highlight the prevalence of significant geographic variation in children malnourished at the level of district, which was then estimated for smaller geographic level, i.e., village-level [17].

Also, there is voluminous research that studied and established significant association between factors such as birth weight and maternal height/age, child's age and sex, family size, illness, and income level, sanitation facilities, early pregnancy, institutional delivery, skilled birth attendance, antenatal and postnatal care and other correlates [18–22] which determine the nutritional status among children under five years of age.

Despite this gamut of research evidence on undernutrition and a wide range of programs, schemes and services offered by the government could not achieve a greater impact on the nutrition status of children [23]. This is also due to the data gap—majorly scarcity of data at a granular level. This reiterates the urgent need for disaggregated data which is critical in strengthening the ground-level interventions. The geotagged data and related analytics such as geospatial and big-data analytics carry tremendous scope in leveraging technology-driven innovations. Hence, the current study has considered the above mentioned socio-economic, demographic and health-related pointers of undernutrition among children and examined these through a geospatial lens, using, for the first time, a block-level data.

Fortuitously, Odisha happens to have carried out a detailed survey at the block level termed Concurrent Child Monitoring Survey (Round-II). Although the survey covered only rural areas and focussed on weight for age of the child as the sole surrogate for nutritional status, it covered all 314 blocks of the state and incorporated many important correlates of child nutrition. Nevertheless, the CCM II data from Odisha available at the ICDS project level is perhaps the only credible sub-district level survey data that facilitates a micro-level analysis, as we will see below. This includes both statistical as well as geospatial analyses.

Hence, the rationale for this study is manifold. At the outset, as mentioned earlier, none of the studies have provided spatial insights at the sub-district level, mainly for want of data. This is inadequate for understanding the variations and nuances at a micro-level, i.e., at the sub-district level. The present study aims to bridge this gap by conducting spatial analysis for a state using the block as a spatial unit and further identifies the correlates of the children

undernourished and their impact, specific to the clusters of high-low burden. And to the best of our review, this methodology of a micro-level, region-specific analytic approach, for evidence-based planning, has been conducted for the first time in the Indian context.

## Data and methods

The nutritional status of children in Odisha shows considerable reduction across the NFHS survey periods. In the four-years between NFHS-4 and the latest NFHS-5 survey periods, the prevalence of children stunted, wasted and underweight has only marginally reduced from 34%, 20% and 34% to 31%, 18% and 30% [4], respectively,—still falling in the 'critical' zone.

The first part of the study explores spatial heterogeneity of child undernutrition in Odisha at the block level, identifies statistically significant clusters with high levels (hot-spots) or low levels (cold-spots) of undernutrition, as well as spatial outliers. The second part of the study, the NSS region-specific analysis, examines differences in socio-economic determinants of undernutrition in three NSS regions of Odisha: Coastal, southern and northern regions. NSS has classified the natural regions based on the natural topography, agro-climatic parameters, and homogeneity of clusters [24]. Within India, there is a low appreciation of the relevance of NSS regions, but this natural region-wise break up can prove valuable, and can highlight inter-regional inequalities within the state, as well as resemblances between contiguous regions across the state.

### Data

The block-wise spatial analysis for Odisha was carried out using the Concurrent Child Monitoring Survey (Round-II), for the year 2014–15 by the Government of Odisha. This dataset is the first in India to provide household data at block level disaggregated by socio-demographic characteristics including caste and standard of living. No further rounds of survey are conducted till now. The data on critical indicators such as health, nutrition, sanitation, and hygiene were collected through household and Frontline Health Worker surveys. CCM-II, covering 4,81,611 households from all 314 blocks for the entire 30 districts of Odisha, is one of the largest state-wide household surveys with a response rate of 90.6% [25]. The CCM-II (2014–15) aggregate level data for Odisha utilised in the study is used with approval in principle by W&CD & Mission Shakti Department, Government of Odisha. This survey received ethical approval from the Indian Institute of Public Health, as mentioned in their technical report [25].

The emergence of clear, strong clusters that happen to be contiguous regions having similar geographical features, rural population densities and crop patterns, from first part of the study laid the rationale for a region-specific unit-level study of socio-economic correlates of undernutrition specific to these identified clusters or the regions classified by the NSSO. Hence, NSS natural region-based study was conducted using the NFHS-4 unit-level dataset due to the non-availability of the CCM II unit-level data. The NFHS-4 child-wise data was used to establish differences in estimates within the state and to show how the emerging clusters perform with respect to children underweight and its correlates.

The NFHS-4 unit-level data is available in the public domain and was accessed from the repository of Demographic Health Survey (DHS) at www.dhsprogram.com/data/. NFHS-4 adopted a multistage stratified sampling design. The data for Odisha state was extracted from the children's file (IAKR74FL) of NFHS-4 data. The information was collected from 30,242 households, 33,721 women aged 15–49 and 4,634-men aged 15–54.

Districts of Odisha were combined as per NSS natural region composition, grouping districts that were contiguous and with similar geographical topographies, rural population densities and crop-pattern into three regions: coastal, northern and southern [24].

## Variable description

The outcome variable studied was children under five years underweight, i.e., weight-for-age below minus two standard deviations. In order to use a single and comprehensive indicator, underweight has been used, which captures both stunting and wasting (i.e., children who are stunted and wasted are likely to be underweight as well) [26] and based on growth standards defined by WHO [27]. Children underweight and children undernourished have been used interchangeably in the paper.

The various independent variables were chosen based on data availability and the literature review mentioned in the previous section. Table 1 provides a description of study variables.

**Table 1. Description of study variables.**

| CCM-II block-level variables | Description |
|---|---|
| *Outcome variable* | |
| Children underweight (%) | Proportion of children aged below five years who were underweight (low weight-for-age) |
| *Predictor variables* | |
| Sanitation facilities (%) | Proportion of households having improved sanitation facilities |
| Early pregnancy (%) | Proportion of women who have had their first pregnancy before nineteen years. |
| Institutional delivery (%) | Proportion of deliveries that occurred in a health facility |
| Skilled birth attendance (%) | Presence of a skilled birth attendant for home delivery |
| Antenatal care (%) | Proportion of mother's who have had more than four ANCs |
| Postnatal care (%) | Proportion of women received postnatal care within 48 hours of delivery. |
| **NFHS4 Unit level variables** | **Description** |
| *Outcome variable* | |
| Children underweight | Non-underweight = 0, Underweight = 1 |
| *Predictor variables* | |
| Age of the child | Age of the child in months. Categorised as age groups of 0–5, 6–11,12–15,16–18,19–23,24–35,36–47 and 48–59 months. |
| Type of residence | Urban/Rural |
| Sex of the child | Male/female |
| Wealth index | Divided into five categories: 'poorest', 'poor', 'middle', 'richer' and 'richest'. |
| Mother's highest educational level, | Mother's highest educational attainment variable; categorised into four groups: 'no education, 'primary', 'secondary' and 'higher'. |
| Mother's current age | Categorised into: below-20, 20–24, and 25 & above years |
| Birth order number | Categorised into: 1, 2, 3, and 4 & above |
| Birth interval | Preceding birth interval categorised into less than 12-months, 12–23, 24–35, and 36&above months |
| Number of ANC checkups | Categorised into: No ANC, less than 3ANC and four or more ANC |
| Social group affiliation | Social groups categorised as 'scheduled-tribes (ST), scheduled-castes (SC), other backward classes (OBC), and Others |
| Type of toilet facility | Improved/Not Improved |
| Nutritional status of mother | BMI of the mother 18.5 kg/m2 or more, and BMI less than 18.5 kg/m2 |
| Place of delivery | Institutional/Non-Institutional delivery |

## Methods

The first part of the study attempted to understand spatial distribution and autocorrelation of indicators of child undernutrition with the help of various spatial statistical techniques.

**Spatial autocorrelation.** Positive spatial autocorrelation is the phenomenon where a random variable with high or low values tends to cluster in space and when a location is surrounded by neighbours with dissimilar values, is called negative spatial autocorrelation. A spatial weight matrix was computed based on queen contiguity (shared vertices) by giving a value of 1 to neighbours and 0 to non-neighbours. The value of each neighbouring location was multiplied by spatial weight, and then the products were summed- thus calculating the spatial lag variable.

*Moran's* **statistics.** Moran's *I* measure the degree of spatial autocorrelation in a dataset across the geographical unit (sub-district, in the current study). It ranges from -1 to 1; positive Moran's *I* indicate observations with similar values surrounding each other (spatial clusters), and negative Moran's *I* denote observations with high values surrounded by low values or vice-versa (spatial outliers).

Univariate Moran's *I* [28] is defined as

$$I = \frac{N}{\sum_i \sum_j w_{ij}} \frac{\sum_i \sum_j w_{ij}(X_i - \bar{X})(X_j - \bar{X})}{\sum_i (X_i - \bar{X})^2} \tag{1}$$

Where,
$N$ = number of spatial units indexed by *I* and *j*;
$X$ = study variable; $\bar{X}$ = mean of *X*; and
$w_{ij}$ = element of a matrix of spatial weights
Bivariate Moran's *I* is defined as

$$I = \frac{N}{\sum_i \sum_j w_{ij}} \frac{\sum_i \sum_j w_{ij}(X_i - \bar{X})(Y_j - \bar{Y})}{\sum_i (Y_i - \bar{Y})^2} \tag{2}$$

Where,
$N$ = number of spatial units indexed by *I* and *j*;
*X and Y* = study variables; $\bar{X}$ *and* $\bar{Y}$ = mean of *X and Y, respectively*;
$w_{ij}$ = element of a matrix of spatial weights

The Univariate and bivariate LISA (Local Indicators of Spatial Association) statistics were applied to measure the degree of spatial non-stationarity and clustering pertaining to the neighbourhood values [28]. In different shades of green, the LISA significance map displays the locations with significant local Moran statistics [29]. The LISA cluster map illustrates significant locations colour coded by types of spatial autocorrelation. Univariate LISA maps were used to show the geographical clustering across the blocks of different variables. The bivariate LISA described the relation between binary variables, one of the given location and another-the average of its neighbours. The average for the neighbours is denoted by the Y-variable, called the spatial lag variable. Thus, the correlation between predictor variables and weighted average of children underweight (dependent variable) was measured using Bivariate LISA maps.

The software package GeoDa with 999 permutations and a pseudo p-value for cluster of <0.05 specified was used for the spatial analyses.

**Spatial regression.** Spatial models were used to explore and examine spatial relationships, which could help describe factors behind the observed spatial patterns.

The presence of spatial autocorrelation in its error term for the outcome variable was confirmed by the Ordinary Least Square (OLS) regression model; hence, two other spatial regression models: spatial lag model (SLM) and Spatial Error Model (SEM), were applied further.,

The multiple linear regression model can be expressed as:

$$Y_i = \alpha + \beta_1 A_i + \beta_2 B_i + \beta_3 C_i + \beta_4 D_i + \beta_5 E_i + \beta_6 F_i \tag{3}$$

Here,

$Y_i$ = percentage of children underweight in the $i^{th}$ block; $\alpha$ = intercept; $\beta$ = regression coefficient for $j^{th}$ variable where $j$ = 1 (3) 6; $i$ (1, 2,. . ., 314) = no of blocks; A = percentage of women who have had their first pregnancy below 19 years of age in the $i^{th}$ block; B = Percentage of Institutional Delivery in the $i^{th}$ block; C = Percent households with improved sanitation in the $i^{th}$ block; D = Percentage of women who have had more than 4 ANC checkups in the $i^{th}$ block; E = Percentage of skilled births in the $i^{th}$ block; F = Percentage of women received postnatal care within 48 hours of delivery in the $i^{th}$ block [16].

The spatial autoregressive models are the regressions models which account for spatial autocorrelation in the dataset. The conditional autoregressive model (CAR), the simultaneous autoregressive model (SAR), and the moving average (MA) process model, are the three spatial autoregressive models commonly referred in the literatures. Two cases of the simultaneous autoregressive (SAR) models include the spatial lag model (SLM), and the spatial error model (SEM), which were utilised in the current study to model the variations.

The S*patial Lag Model* (SLM) runs under one of the assumptions that neighbouring areas have an effect on the dependent variable's observations, and can be written as follows:

$$Yi = \delta \sum_{j \neq 1} Wij \; Yj + \beta Xj + \varepsilon j \tag{4}$$

Here,

$Y_i$ = prevalence of children underweight for $i^{th}$ block; $\delta$ = spatial autoregressive coefficient; $W_{ij}$ = spatial weight of proximity between block $i$ and $j$; $Y_j$ = prevalence of children underweight in $j^{th}$ block; $\beta_j$ = coefficient; $X_j$ = predictor variable and $\varepsilon_i$ = residual.

While in the *Spatial Error Model (SEM)*, the effect of missing variables (in the model) on the outcome variable is considered. The spatial error model considers spatial dependence in the error term, unlike the spatial lag model [30]. SEM is stated as follows:

$$Y_i = \beta X_j + \lambda \sum_{j \neq 1} W_{ij} \; Y_j + \beta X_j \varepsilon_j + \varepsilon_i \tag{5}$$

Here,

$Y_i$ = prevalence of children underweight for $i^{th}$ block; $\lambda$ = spatial autoregressive coefficient; $W_{ij}$ = spatial weight of proximity between block $i$ and $j$; $Y_j$ = prevalence of children underweight in $j^{th}$ block; $\beta_j$ = coefficient; $X_j$ = predictor variable and $\varepsilon_i$ = residual [16].

**Logistic regression.** The second part of the study conducted an NSS natural region-specific analysis using NFHS-4; bivariate and multivariate analysis, in the form of cross-tabulations and binary logistic regressions, respectively, were applied. Adjusted odds ratio was estimated to evaluate the association between a child's individual factors, maternal, socio-economic and demographic factors with prevalence of undernutrition. STATA version 14 (StataCorp™, Texas) was used to conduct the unit-level analysis.

## Results

### Spatial heterogeneity of children underweight and its correlates

The descriptive statistics of CCM II block-level data has been presented in Table 2. All the selected dependent and independent variables distribution displayed wide variation across the blocks of Odisha.

The prevalence map illustrates the spatial pattern of underweight as per the CCM II dataset (Fig 1). Green stands for low prevalence and red for high prevalence. NSS natural region-wise boundary map was superimposed to highlight the three different patterns that emerge within the state. According to WHO benchmarks, undernutrition in rural Odisha is 'critical' and is highly significant. As per CCM-II, the primary outcome variable- children underweight, had a high prevalence across the blocks of Odisha (43.8%). The state's coastal belt performed relatively better and had children underweight prevalence of less than 38%, whereas northern and southern blocks of the state performed very poor with a prevalence of more than 48% (130 blocks out of 314). Similar clustering was observed for all selected correlates (S1 File); the coastal belt performed well as compared to northern and southern regions.

The corresponding LISA maps (Figs 2 and 3) indicate that while significant clusters of high-high spatial association were shown for children underweight in 55 blocks located in southern and northern regions, 70 blocks in the coastal region showed a significant low-low spatial association. Other LISA results can be found in the S1 File. Statistically significant clusters falling in the three NSS natural regions of the state-directed that overall, coastal region performed well with respect to all the studied parameters, as compared to northern and southern regions.

Statistically significant Moran's *I* values were observed for all selected variables (Table 3). Moran's *I* value for children underweight was pretty strong at 0.73. Proportion of women having early pregnancies showed strong spatial autocorrelation, with a Univariate Moran's *I* value of 0.68, followed by Institutional delivery and skilled birth attendance (Moran's *I*: 0.65, pseudo p<0.001).

The Bivariate LISA maps for underweight against correlates are presented Figs 4 and 5 (other results in S1 File). Correlates like improved sanitation, early pregnancy, institutional delivery, skilled births, antenatal care and postnatal care showed significant spatial autocorrelation with children underweight.

A significant positive spatial autocorrelation was seen between underweight and women having first early pregnancy, with Moran's *I* value = 0.43. A strong negative spatial autocorrelation was observed between underweight and all other correlates, namely sanitation conditions (Moran's *I*: -0.60), institutional delivery and skilled births (Moran's *I*: -0.38), woman who had more than four ANCs (Moran's *I*: -0.30) and received postnatal care (Moran's *I*: -0.35) (Table 3).

**Table 2. Summary statistics of the selected dependant and independent variables.** Block-level percentages, CCMII, 2014–15, Odisha, India.

| Variable | Mean | Std. Dev. | Min | Max |
|---|---|---|---|---|
| Children Underweight | 43.8 | 11.0 | 16.7 | 68.2 |
| Institutional Delivery | 82.2 | 15.8 | 6.3 | 100 |
| First pregnancy ≤19 years of age | 31.2 | 13.2 | 6.5 | 67.3 |
| More than 4 ANCs | 47.2 | 21.0 | 1.9 | 100 |
| Households having improved sanitation facilities | 12.9 | 10.6 | 0 | 47.3 |
| Women received postnatal care within 48 hours of delivery | 90.3 | 12.4 | 21.4 | 100 |
| Skilled Attendance at Birth | 83.6 | 15.4 | 12.5 | 100 |

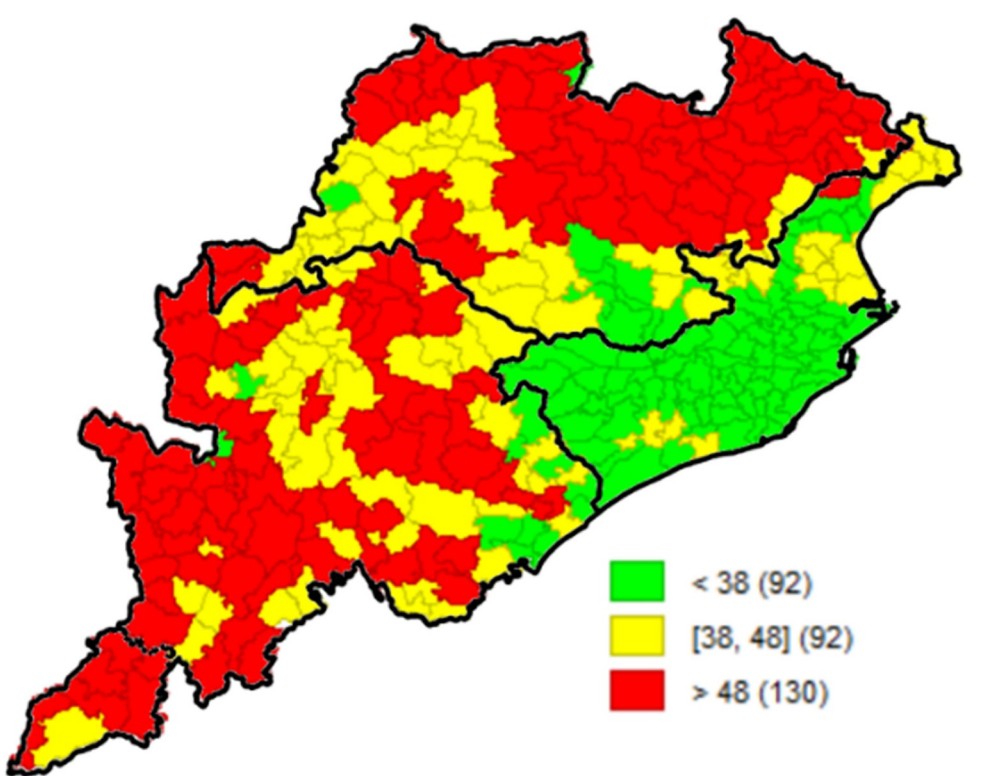

**Fig 1. Prevalence of children underweight, block-wise, CCM-II, 2014–15, Odisha, India.**

Bivariate LISA cluster map of underweight with sanitation (Fig 4) indicated that about 64 of 314 blocks located at coastal region had the lowest prevalence of underweight and highest level of improved sanitation. And about 53 of 314 blocks situated in southern and northern regions had high underweight prevalence and households with low sanitation. Significant clusters were found to have high children underweight prevalence but had high improved sanitation (4 blocks), and clusters with low underweight prevalence but had lack of sanitation facilities (6 blocks). With respect to children underweight and woman having early pregnancy (Fig 5), significant clusters of blocks having high underweight prevalence and high prevalence of women having pregnancy below 19 years of age were found (44 blocks), located in northern and southern regions, and 54 cold-spot clusters were found in the coastal region.

Other bivariate LISA analyses indicate that coastal region in general performed well, having blocks with low underweight prevalence and high institutional deliveries, high skilled birth attendance, high ANCs and postnatal care. On the contrary, northern and southern regions performed poorly, having blocks with a low prevalence of institutional deliveries, skilled birth attendance, ANCs and postnatal care, and hence had a high prevalence of underweight children. Thus, all these findings give us a better understanding of regional level patterns across the blocks, on undernutrition among children than findings at a state or district level.

## Spatial regression- children underweight and its correlates

Spatial regression models were applied to examine spatial relationships describing factors underlying the spatial patterns. The results of the OLS regression model are illustrated in Table 4. This showed an initial check of association between the outcome variable, i.e., children underweight, and its correlates at the block level, devoid of the spatial structure of the data.

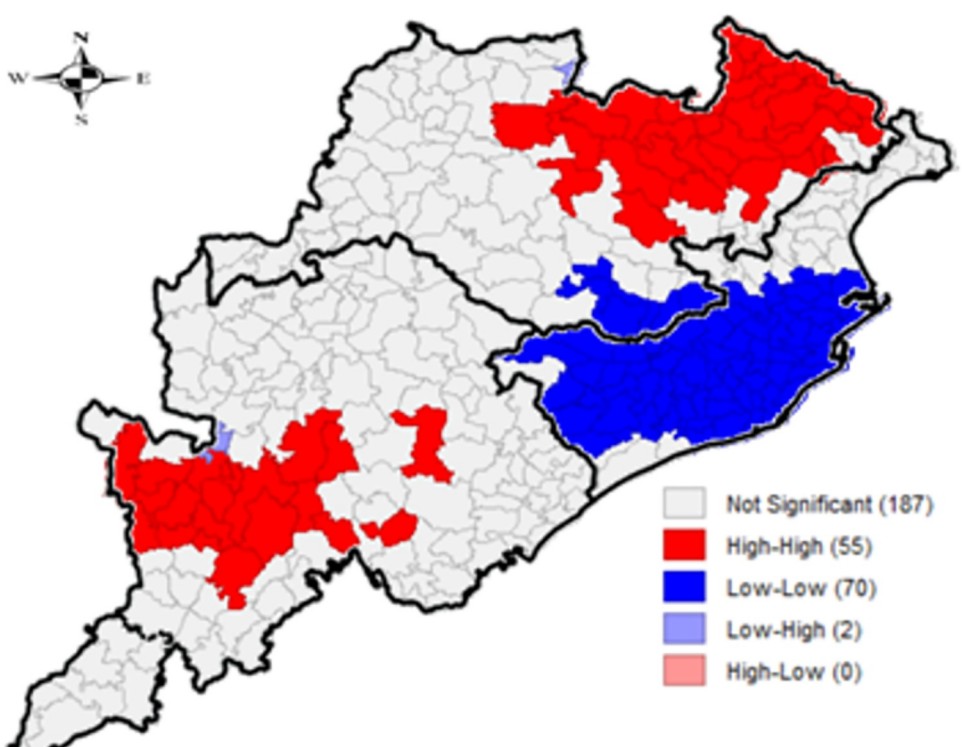

**Fig 2. Univariate LISA cluster map for children underweight, block-wise, CCM-II, 2014–15, Odisha, India.**

The diagnostics of the OLS model showed that the data was normally distributed, but the results indicate spatial autocorrelation in the residuals of the model, i.e., underweight: Moran's $I = 0.42$, pseudo p-value$< 0.000$, and also both simple tests of the Lagrange Multiplier (lag and

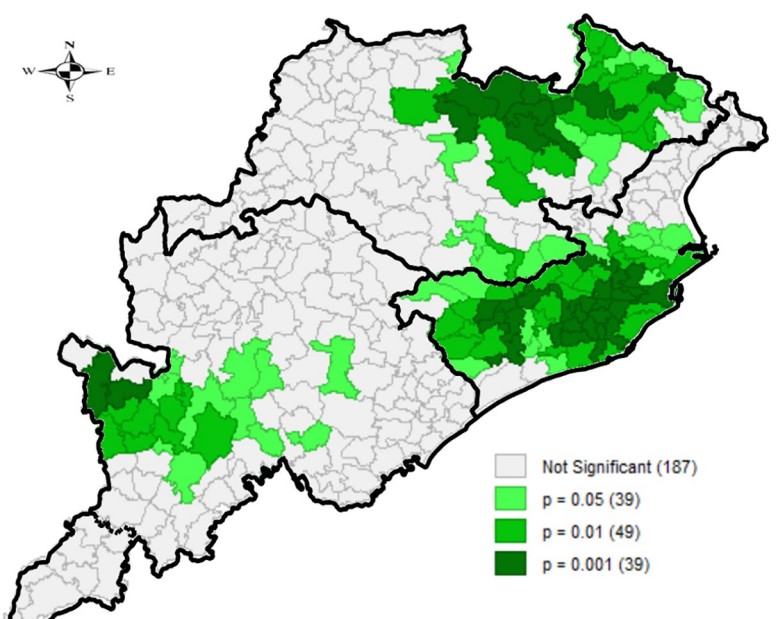

**Fig 3. Univariate LISA significance map for children underweight, block-wise, CCM-II, 2014–15, Odisha, India.**

**Table 3. Univariate and bivariate Moran's *I* statistics of the prevalence of children underweight and its micro-scale correlates in Odisha, CCM-II, 2014–15.**

| Micro-scale variables (block-level percentages) | Univariate Moran's *I* | Bi-variate Moran's *I* |
|---|---|---|
| Children Underweight | 0.73 | - |
| Institutional Delivery | 0.65 | -0.38 |
| Skilled Attendance at births | 0.65 | -0.38 |
| First pregnancy ≤19 years of age | 0.68 | 0.43 |
| Women received postnatal care within 48 hours of delivery | 0.57 | -0.35 |
| Households having improved Sanitation | 0.64 | -0.60 |
| More than 4 ANCs | 0.45 | -0.30 |

All Moran's *I* value significant (pseudo p<0.001) at 999 permutations

error), were significant, indicating the presence of spatial dependence. This suggested non-uniform distribution of prevalence of underweight among children across blocks of Odisha and their occurrence in specific clusters. Therefore, the absence of spatial dependency taken as the null hypothesis was rejected, and the alternate hypothesis of the occurrence of positive spatial autocorrelation with respect to underweight was thus considered. Given the presence of spatial autocorrelation, spatial regression models were estimated further.

The spatial effect was modelled in two ways by estimating spatial lag and spatial error model. While controlling for spatial dependence, the models were re-estimated with the maximum likelihood approach. Estimates of both SEM and SLM confirmed that three correlates: women having more than four ANC visits, early pregnancy and households with improved sanitation, same as the OLS model results, remained the statistically significant predictors of children underweight (Table 4).

Based on the model diagnostics, the SEM was found to be an improved fit, having greater R-squared and Log-likelihood values and a smaller AIC. The final and spatial endogeneity adjusted estimates of correlates for the prevalence of children underweight across the blocks were derived from the results of the SEM model [31].

With respect to children underweight, the largest statistically significant coefficient was for household sanitation (β = -0.33), followed by women having an early pregnancy and ANC visits (β = −0.11). The coefficient estimated for improved household sanitation confirmed a ten-point

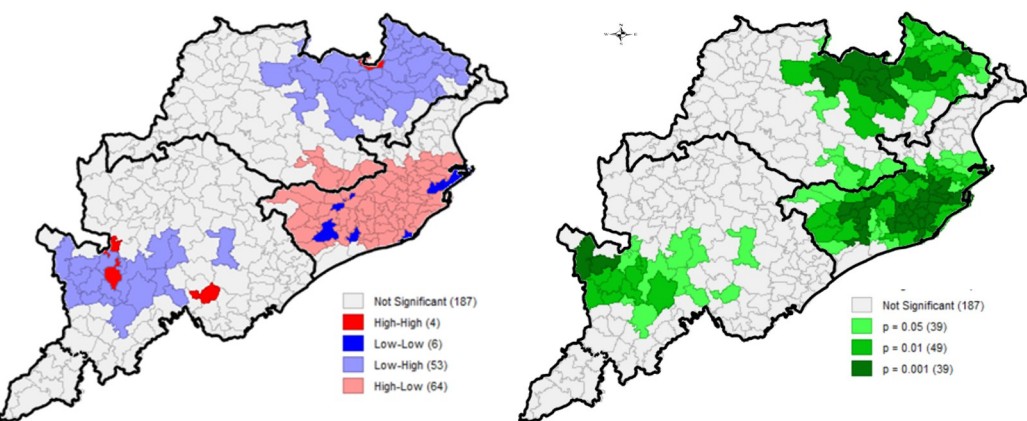

**Fig 4. Bivariate LISA cluster and significance map of underweight vs improved sanitation, block-wise, CCM-II, 2014–15, Odisha, India.**

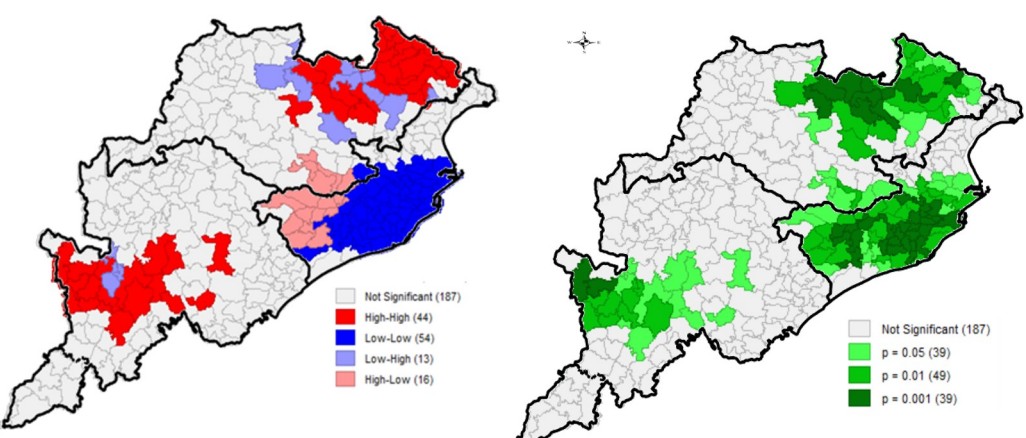

**Fig 5. Bivariate LISA cluster and significance map of underweight vs first pregnancy ≤19 years of age, block-wise, CCM-II, 2014–15, Odisha, India.**

increase in the proportion of household sanitation across the blocks, was associated with a 3.3-point reduction in the prevalence of children underweight. On the other hand, a 10-point increase in the percentage of early pregnancies led to a rise in the children underweight prevalence by 1.1 units. A ten-point increase in the percentage of women having more than four ANC visits was supposed to reduce the percentage of children underweight by 1.1 points.

A highly significant value of the lag coefficient on spatially correlated errors (LAMBDA) of 0.77 was observed. It indicates that one-unit change in the error in the neighbouring blocks caused a 0.77-unit change in children underweight, i.e., any change in the omitted variables which could have an effect on the children underweight prevalence, but were not existing in the study, also could lead to a change in the children underweight prevalence in the neighbourhood blocks [16, 32, 33].

## Region-specific study: NFHS-4 unit level analysis

From the NSS boundary shapefiles superimposed onto block-wise shapefiles, the occurrence of clear spatial clusters was found. A clear coastal and non-coastal regional divide was observed

**Table 4. Results- spatial regression models: OLS, spatial lag & spatial error model, block-wise, CCM-II, 2014–15, Odisha, India.**

| Block-level Correlates | OLS | | Spatial Lag Model | | Spatial Error Model | |
|---|---|---|---|---|---|---|
| | Coefficient | Probability | Coefficient | Probability | Coefficient | Probability |
| Institutional Delivery | -0.137 | 0.610 | 0.025 | 0.899 | 0.130 | 0.530 |
| First pregnancy ≤19 years of age | 0.165 | **0.000** | 0.058 | 0.048 | 0.116 | 0.003 |
| Skilled births | 0.235 | 0.406 | 0.062 | 0.767 | -0.124 | 0.554 |
| Households having improved Sanitation | -0.652 | **0.000** | -0.315 | **0.000** | -0.332 | **0.000** |
| More than 4 ANCs | -0.094 | **0.000** | -0.073 | **0.000** | -0.114 | **0.000** |
| Women received postnatal care within 48 hours of delivery | -0.036 | 0.763 | -0.049 | 0.581 | 0.003 | 0.969 |
| R- squared value | 0.579 | | 0.765 | | 0.777 | |
| Adjusted R square | 0.570 | | | | | |
| Lambda Value (Lag Coefficient) | | | | | 0.773 | 0.000 |
| Rho Value (Lag coefficient) | | | 0.629 | 0.000 | | |
| Log likelihood | | | -988.602 | | -989.036 | |
| AIC value | 2141.37 | | 1987.20 | | 1986.07 | |
| No. of blocks | 314 | | 314 | | 314 | |

from the spatial analysis results, showing coastal region performed well for underweight children and its correlates as compared to northern and southern regions. The prevalence of children underweight was observed highest in the northern region at 40.6%, followed by the southern region having 39.5% prevalence, and the lowest prevalence was seen in the coastal region at 23.6%. This prompted a region-specific study to examine how the correlates impact undernutrition condition, cluster-specific.

The population distribution according to socio-economic indicators, as per NSS natural regions of Odisha, and bivariate analysis results in the form of cross-tabulation and Pearson's chi-square are presented in S2 File.

The binary logistic results are illustrated in Fig 6 & Table 5. Model 1 (Fig 6) describes the computed odds ratio for children underweight by taking their location as per NSS natural regions along with their socio-demographic background characteristics by binary logistic regression. The main determinants of focus in this study, i.e., three regions of the state, southern, northern, and coastal regions, had a statistically significant effect on the prevalence of undernutrition after controlling the socio-economic, maternal factors.

When all factors were controlled, it was seen that children under-five years of age from the coastal region were less likely to be underweight than children from the southern and northern regions of the state. That is, children from the southern region had a 55% higher chance of being underweight than children in the coastal region. Similarly, as compared to the coastal region, children from the northern region had a 74% higher chance of being underweight.

It can also be interpreted from these results that, when controlled for differences in regions within the state of Odisha, correlates like age and sex of the child, birth order number, birth interval, wealth quintile, mother's education level, nutritional status of the mother, ANC visits had a statistically significant impact on nutritional status of the child. Children aged 12–15 months were 1.16 times more likely to be underweight when compared to 0–5 months aged children, indicating significant growth faltering. As the wealth quintile of the respondent increased from poorest to richest, the odds of the child becoming undernourished decreased. It was also noticed that maternal education also significantly affected underweight among

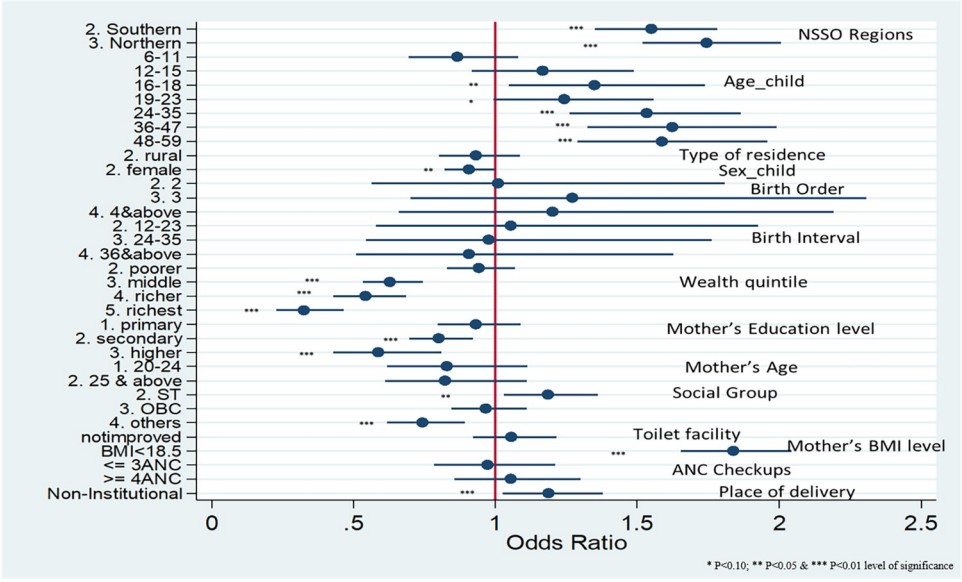

**Fig 6. Model 1: Impact of NSSO regions, along with socio-economic background characteristics, on children underweight, for Odisha, assessed by logistic regression using NFHS-4 (2015–16).**

**Table 5. Impact of the background, child and maternal characteristics, on the child underweight as per the Odisha NSSO natural-regions, assessed by adjusted logistic regression, NFHS-4 (2015–16).**

| Background, Child and Maternal Characteristics | Coastal (Model 2) | Southern (Model 3) | Northern (Model 4) |
|---|---|---|---|
| | AOR with 95% C.I | AOR with 95% C.I | AOR with 95% C.I |
| **Age of the child in months** | | | |
| 0–5Ⓡ | 1 | 1 | 1 |
| 6–11 | 0.593**(0.36,0.96) | 0.978 (0.706,1.354) | 0.94 (0.638,1.383) |
| 12–15 | 0.77 (0.455,1.301) | 1.397*(0.978,1.995) | 1.222 (0.799,1.869) |
| 16–18 | 0.626 (0.352,1.114) | 1.648***(1.14,2.39) | 1.753**(1.115,2.75) |
| 19–23 | 0.604**(0.37,0.99) | 1.656***(1.19,2.31) | 1.354 (0.911,2.012) |
| 24–35 | 0.835 (0.55,1.266) | 1.977***(1.47,2.65) | 1.62***(1.15,2.28) |
| 36–47 | 1.294 (0.849,1.97) | 2.027***(1.49,2.76) | 1.381*(0.965,1.97) |
| 48–59 | 1.036 (0.671,1.6) | 2.104***(1.53,2.88) | 1.435*(0.993,2.07) |
| **type of place of residence** | | | |
| urbanⓇ | 1 | 1 | 1 |
| Rural | 1.071 (0.774, 1.481) | 1.075 (0.803, 1.439) | 0.791**(0.628, 0.996) |
| **wealth index** | | | |
| poorestⓇ | 1 | 1 | 1 |
| Poorer | 0.801 (0.611, 1.051) | 0.964 (0.801, 1.161) | 1.014 (0.803, 1.279) |
| Middle | 0.536***(0.389, 0.738) | 0.604***(0.463, 0.788) | 0.68**(0.505, 0.917) |
| Richer | 0.476***(0.306, 0.741) | 0.453***(0.297, 0.691) | 0.629**(0.423, 0.937) |
| Richest | 0.333***(0.15, 0.739) | 0.187***(0.089, 0.392) | 0.386***(0.227, 0.656) |
| **sex of child** | | | |
| maleⓇ | 1 | 1 | 1 |
| Female | 1.04 (0.844, 1.282) | 0.882*(0.76, 1.024) | 0.866*(0.731, 1.026) |
| **Social Group** | | | |
| SCⓇ | 1 | 1 | 1 |
| ST | 1.402 (0.936, 2.098) | 1.13 (0.925, 1.381) | 1.448***(1.132, 1.851) |
| OBC | 0.683***(0.521, 0.893) | 1.121 (0.908, 1.384) | 1.097 (0.847, 1.42) |
| Others | 0.676***(0.506, 0.902) | 1.123 (0.796, 1.585) | 0.556***(0.386, 0.8) |
| **Type of toilet facility** | | | |
| ImprovedⓇ | 1 | 1 | 1 |
| Not Improved | 1.155 (0.878, 1.518) | 1.07 (0.848, 1.35) | 0.932 (0.738, 1.176) |
| **birth order number** | | | |
| 1Ⓡ | 1 | 1 | 1 |
| 2 | 1.599 (0.49, 5.219) | 1.457 (0.475, 4.472) | 0.614 (0.255, 1.478) |
| 3 | 2.163 (0.642, 7.289) | 1.985 (0.642, 6.135) | 0.677 (0.272, 1.687) |
| 4 & above | 1.523 (0.432, 5.368) | 2.139 (0.689, 6.638) | 0.592 (0.235, 1.494) |
| **preceding birth interval (months)** | | | |
| <12Ⓡ | 1 | 1 | 1 |
| 12–23 | 0.645 (0.188, 2.213) | 0.506 (0.162, 1.584) | 2.573**(1.03, 6.425) |
| 24–35 | 0.766 (0.229, 2.559) | 0.489 (0.159, 1.509) | 1.924 (0.789, 4.691) |
| 36&above | 0.715 (0.218, 2.349) | 0.444 (0.145, 1.361) | 1.862 (0.774, 4.476) |
| **Mother's highest educational level** | | | |
| no educationⓇ | 1 | 1 | 1 |
| Primary | 0.841 (0.578, 1.224) | 1.014 (0.813, 1.266) | 0.778*(0.582, 1.041) |
| Secondary | 0.717*(0.511, 1.006) | 0.853 (0.696, 1.046) | 0.747**(0.581, 0.96) |
| Higher | 0.267***(0.122, 0.584) | 0.888 (0.494, 1.597) | 0.593**(0.365, 0.963) |
| **Mother's current age** | | | |

*(Continued)*

**Table 5.** (Continued)

| Background, Child and Maternal Characteristics | Coastal (Model 2) | Southern (Model 3) | Northern (Model 4) |
|---|---|---|---|
| | AOR with 95% C.I | AOR with 95% C.I | AOR with 95% C.I |
| below 20Ⓡ | 1 | 1 | 1 |
| 20–24 | 0.629 (0.32, 1.238) | 0.793 (0.521, 1.207) | 1.105 (0.656, 1.861) |
| 25 & above | 0.513*(0.259, 1.016) | 0.874 (0.564, 1.353) | 1.107 (0.652, 1.882) |
| **Nutritional Status of Mother** | | | |
| BMI> = 18.5Ⓡ | 1 | 1 | 1 |
| BMI<18.5 | 1.743***(1.368, 2.221) | 1.823***(1.558, 2.134) | 1.921***(1.594, 2.315) |
| **No. of ANC visits** | | | |
| No ANCⓇ | 1 | 1 | 1 |
| < = 3ANC | 0.863 (0.571, 1.304) | 1.146 (0.811, 1.62) | 0.886 (0.6, 1.308) |
| > = 4ANC | 1.03 (0.69, 1.538) | 1.251 (0.896, 1.745) | 0.934 (0.643, 1.358) |
| **Place of Delivery** | | | |
| InstitutionalⓇ | 1 | 1 | 1 |
| Non-Institutional | 1.002 (0.659, 1.522) | 1.258**(1.037, 1.526) | 1.288*(0.962, 1.725) |

\* P<0.10

\*\* P<0.05

\*\*\* P<0.01 level of significance

children; mothers who were highly educated had 0.58 lesser odds of their child being underweight compared to mothers with no education. Mothers with BMI levels below normal had 1.83 higher odds of their child being underweight. Non-institutional births had 1.18 times higher odds of the child being underweight.

After studying determinants of undernutrition for the state as a whole, Models 2–4 (Table 5) depict the results of a region-specific study on correlates of malnutrition.

Model 2 illustrates the adjusted odds ratio for various socio-economic factors and their impact on the nutritional status of children for coastal region-specific. Results from this model show that wealth profile, social group, mother's age, education level, and nutritional status; were found to be significantly associated with underweight in children in the coastal region. Model 3 shows that variables like age and sex of the child, wealth profile, nutritional status of the mother, and place of delivery had a significant effect on underweight children in the southern region. Evident growth faltering was noticed; children aged 16–18 months were 1.6 times more likely to be underweight than children below-six months of age. Mother's education did not show a statistically significant association in this region. Model 4 depicts the adjusted odds ratio for the northern region. Variables showing significant association with children underweight in this region were majorly age and sex of the child, type of residence, wealth profile, social group, birth interval, mother's education and nutritional level, and place of delivery. Like the southern region, children from the northern region also suffered from growth faltering; 16–18 months' children were 1.7 times more likely than children below six months to be underweight. Children from a tribal population in this region were 44% more likely to be underweight.

## Discussion

The geo-analytic techniques helped identify and visualise new data patterns and connections, often bringing the unseen to the forefront through spatial analyses facilitating new insights

and hypotheses. Even though many studies are available at the macro and meso-level, studies conducted on undernutrition among children in India at the micro-level are rare, mainly due to the non-availability of micro-level data. The current analysis throws more light on spatial heterogeneity of undernutrition among the children at the block-level for the state of Odisha, thereby helping in context-specific decentralised planning. Below are the noticeable findings from the study.

The primary outcome variable considered in this study, children underweight, had a high prevalence throughout the blocks of rural Odisha. With the help of the superimposed boundary shapefile of NSS natural regions, the occurrence of clear spatial patterns was identified. A clear coastal and non-coastal regional divide was observed from the spatial analysis results, showing better performance of coastal region as compared to northern and southern regions of the state. As compared to the coastal region, the poverty ratio for southern and northern regions was higher. The southern and northern rural regions also comprised of a higher number of tribal populations. This establishes some linkages between poverty and undernutrition in the state as highlighted in earlier studies [34, 35] and distribution of poverty in these regions is reflected in the spatial contours for undernutrition. These findings are also in sync with the chronic poverty differences across the NSSO natural regions established in an earlier study [36], which implies no significant improvement in the economic status even after a decade. This needs more in-depth and detailed research at the NSSO level to study its policy implications.

The spatial association and emergence of significant clusters showing the prevalence of underweight in all three regions indicate a need to focus more on northern and southern blocks. Based on the findings, it also necessitates the need to develop and implement different approaches or models which can eradicate child undernutrition in these areas. These models can also be used as best practices for other high burdened regions.

For maternal health indicators such as institutional deliveries, skilled birth attendance, women having more than four ANC and receiving postnatal care within 48 hours of delivery, blocks located in the southern region of the state performed very poorly, followed by northern region blocks. Also, these showed significant spatial autocorrelation with children underweight. This establishes findings from earlier studies showing the impact of maternal health on children underweight [37, 38] and emphasises the strong need for addressing issues related to antenatal and postnatal care and services.

Early pregnancy and child undernutrition are interconnected social and public health concerns globally and more significant in the case of Indian states, where the prevalence of early pregnancy is high [39, 40]. The current study findings with respect to children underweight and early pregnancies showed significant clusters of blocks having high underweight prevalence and high prevalence of women having pregnancy below 19 years of age in northern and southern regions of the state, and cold-spots clusters were found in blocks of the coastal region. This validates that adolescent pregnancies have a negative impact on child health outcomes and suggests increased targeted interventions such as increasing age at first marriage, age at first birth, and adolescent centred approaches and policies towards improving maternal and child nutrition.

There is a growing base of evidence showing a significant link between sanitation and child nutritional outcomes [41–43]. The present study showed that blocks in the coastal region of Odisha had the lowest prevalence of underweight and highest level of better sanitation, whereas blocks located in southern and northern regions of the state had high underweight prevalence and households with inadequate or poor sanitation. The Swachh Bharat Abhiyan launched in 2019 aims to address this problem by focussing on improving infrastructural facilities for better sanitation facilities in rural areas. With this in progress, we can expect an

improvement in child nutritional outcomes in the coming years. Also, this indicates the need for multi-sectoral coordination between concerned civic and health departments to address these issues cohesively.

The presence of spatial dependence was observed from the results, suggesting the prevalence of unequal distribution of children underweight across the blocks of Odisha; instead, it occurred in specific clusters. With respect to children underweight, the largest statistically significant impact was observed for household sanitation followed by women having early pregnancies and antenatal care.

Further, the child-wise study based on the NSS natural regions of Odisha highlighted that the three regions, southern, northern and coastal regions, showed a statistically significant effect on the prevalence of undernutrition. When all factors were controlled, it was seen that children below five years of age from the coastal region were less likely to be underweight than children from southern and northern regions, which conforms with results from the spatial analysis. When adjusted for differences in the regions, correlates like age and sex of the child, birth order number, birth interval, wealth quintile, mother's education level and nutritional status, ANC visits had a statistically significant impact on the nutritional status of the child.

In the coastal region, wealth index, mother's education, and nutrition status, ANC visits, showed a statistically significant impact. For the southern region, the odds ratio for socio-economic factors and their impact on nutritional status of children, indicating child's age, birth order number, wealth profile of households, nutritional status of the mother, and place of delivery having a strong impact on undernutrition indicators. For the northern region, age of the child, birth order number, household's wealth quintile, BMI level of mothers, institutional births, ANC visits had an impact on undernutrition indicators. Growth faltering was evident in the non-coastal regions of Odisha. These findings do beg a more nuanced analysis of correlates across natural regions at different levels of development and prosperity, for the parameters requiring attention from the programme implementation point of view may be different across the 'development ladder'.

Determinants like birth order of the child, household wealth profile, household's caste, mother's nutritional status, proved to be the strongest determinants with respect to the state as a whole, i.e., when controlled for the difference in geographical location, and also when looked at the region in isolation. Hence, these are factors that need to be prioritised.

These findings are consistent with results from earlier studies conducted in countries with low-middle-income economies, including India and its states [44–46]. In addition to this, when we speak specifically about Odisha, recent studies confirm very narrow progress in sanitation, women's secondary education, early marriage among girls, and poverty in the state, which are some core determinants of child undernutrition [47, 48]. Even though programmes such as National Nutrition Mission (POSHAN Abhiyan) and National Sanitation Programme (Swatch Bharat Abhiyan) are being carried out by State and Central Governments, still it is essential to identify missing links and for more concentrated efforts to improve the nutritional status of the country. The spatial analysis points towards the need for prioritisation and strengthening of nutritional programmes and interventions in the highlighted geographic enclaves. Also, findings stress the need for identifying cohesive approaches, which is critical to eliminating child undernutrition at the state level.

Additionally, the study points towards the necessity of having good quality data at the micro-level for all the states, which will help in more focussed and context-specific efforts. It is also essential to understand inter-region differences and address risk factors of child undernutrition at the micro-level, which is imperative for the success of local and programs.

The patterns of agriculture, food security, climatic and environmental factors are crucial aspects impacting geographical location and space in different determinants of nutrition,

which could not be addressed in the current study. Integrating agricultural and environmental data in future research could help policymakers develop cohesive strategies for sustainable solutions addressing undernutrition. Also, due to the non-availability of unit level data of the CCM-II dataset, the impact of various demographic and socio-economic factors could not be assessed at the sub-district level using the same dataset. This further points towards the need for collection of more good quality granular data and its dissemination. Another limitation of the study could be related to the two datasets used, both of which are secondary in nature, and could possibly entail measurement error in terms of child anthropometric recordings.

## Conclusion

This study establishes spatial heterogeneity of child undernutrition in rural areas of Odisha. The findings could be beneficial for contextualised and evidence-based planning of focussed intervention programmes and schemes addressing child undernutrition. The spatial clustering of different socio-demographic indicators in specific geographic pockets highlights the differential impact of these determinants on child undernutrition thereby reinforcing a strong need for targeted intervention in these areas. The findings also suggest the need for multi-sectoral and integrated efforts to address poverty, hygiene and sanitation, maternal health and nutrition at the state level.

The study also helps in identifying potential areas which are closer to the goal post and can thus fast track process of creating and upscaling malnutrition-free enclaves. Present findings also stress the necessity to have micro-level data for other states of India to understand undernutrition in a more nuanced way. Given the sheer regional diversity in India, both in terms of child undernutrition and its correlates, it is essential to develop a region-specific evidence base at the micro-level. Thus, help in developing a customised plan of action for each area and to produce undernutrition free regions with a consequent ripple effect and thereby building a more cohesive and sustainable malnutrition-free India.

## Supporting information

**S1 File. Odisha block-wise univariate and bivariate LISA results.**
(PDF)

**S2 File. Differences across NSS regions of Odisha: NFHS-4 unit level analysis.**
(PDF)

**S1 Fig. Flowchart of the data and methods used in the current study.**
(TIF)

## Acknowledgments

An earlier version of this manuscript was presented at the Consortium of Universities for Global Health (April, 2021) and at the International Population Conference, IUSSP (December, 2021). All the comments of the participants helped in the strengthening of this manuscript.

## Author Contributions

**Conceptualization:** Apoorva Nambiar, Satish B. Agnihotri.

**Data curation:** Apoorva Nambiar.

**Formal analysis:** Apoorva Nambiar.

**Methodology:** Apoorva Nambiar.

**Supervision:** Satish B. Agnihotri, Ashish Singh, Dharmalingam Arunachalam.

**Visualization:** Apoorva Nambiar.

**Writing – original draft:** Apoorva Nambiar.

**Writing – review & editing:** Apoorva Nambiar, Satish B. Agnihotri, Ashish Singh, Dharmalingam Arunachalam.

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
