## [Decision Letter · Decision Letter 0]

7 Mar 2022

PONE-D-22-02352Region matters: Mapping the contours of undernourishment among children in Odisha, IndiaPLOS ONE

Dear Dr. Agnihotri,

Thank you for submitting your manuscript to PLOS ONE. After careful consideration, we feel that it has merit but does not fully meet PLOS ONE’s publication criteria as it currently stands. Therefore, we invite you to submit a revised version of the manuscript that addresses the points raised during the review process.

 The manuscript focuses on a topic of interest. However, for its improvement, the reviewer's comments and the following points should be considered.

-Elaboration of a paragraph with the limitations of the study.

-Discussion about possible differences between the period of data collection and the present day.

-Explain that the spatial lag model and Spatial Error Model are the CAR and SAR models.

-Inform the distribution of the data in the spatial regressions. Check if other distributions (eg Poisson or negative binomial) have a better fit.

-Implement or justify the non-implementation of multilevel models.

We look forward to receiving your revised manuscript.

Kind regards,

Vinícius Silva Belo

Academic Editor

PLOS ONE

Journal Requirements:

3. We note that Figures 1-5 and S1 file in your submission contain map/satellite images which may be copyrighted. All PLOS content is published under the Creative Commons Attribution License (CC BY 4.0), which means that the manuscript, images, and Supporting Information files will be freely available online, and any third party is permitted to access, download, copy, distribute, and use these materials in any way, even commercially, with proper attribution. For these reasons, we cannot publish previously copyrighted maps or satellite images created using proprietary data, such as Google software (Google Maps, Street View, and Earth). For more information, see our copyright guidelines: http://journals.plos.org/plosone/s/licenses-and-copyright.

a) You may seek permission from the original copyright holder of Figures 1-5 and S1 file to publish the content specifically under the CC BY 4.0 license.  

Reviewers' comments:

Reviewer's Responses to Questions

**Comments to the Author**

1. Is the manuscript technically sound, and do the data support the conclusions?

Reviewer #1: Yes

Reviewer #2: Yes

Reviewer #3: Yes

2. Has the statistical analysis been performed appropriately and rigorously? 

Reviewer #1: I Don't Know

Reviewer #2: Yes

Reviewer #3: Yes

3. Have the authors made all data underlying the findings in their manuscript fully available?

Reviewer #1: Yes

Reviewer #2: Yes

Reviewer #3: Yes

4. Is the manuscript presented in an intelligible fashion and written in standard English?

Reviewer #1: Yes

Reviewer #2: Yes

Reviewer #3: Yes

5. Review Comments to the Author

Reviewer #1: The article presents important data on child malnutrition in India, with geospatial analysis associated with socioeconomic and health characteristics of the population according to region of residence. Its results are of relevance to the area of Nutrition and Public Health. The article is properly written in formal English and has a clear and direct format.

Reviewer #2: Objective: The aim of the study was to investigate undernutrition among children in Odisha, Indian, at sub-district levels using data from the Concurrent Child Monitoring Survey-II, 2014-15, through a geospatial analysis regional, state, district, urban-rural.

Comments:

This is an important study, of an unprecedented nature in the country, in which this analysis of the spatial distribution of child malnutrition by block or sub-district was carried out for the first time. Therefore, it is aligned with the scope of the journal Plos One.

- The methods are adequately described, allowing the reproduction of the study and providing a clear display for the results.

- The results are properly presented and the discussion dialogues with the scientific literature

- Conclusion: the authors highlight the main findings and contributions of the study to adress public policies to combat child malnutrition in different regions India. Below are some considerations.

- Considering the persistense of undernutrition as a public heath problem in Índia, and the originality of this study, the manuscript could be eligible for publication in Plos One, after a few changes listed below:

• Abstract: In order to facilitate the communication with the cientific readers, I suggest the authors to build a structured Abstract: Background; Objective; Methods; Results; Conclusion.

• Introduction: I suggest shifting the items: “Data and Methods, Data, Variable Description” to the first paragraph of the methods section. Building a flowchart could help improving the concisenes of the Methods.

• Figures and Tables: It is necessary to supply complete titles and legends for all figures and Tables, including the country and the year of the study.

Reviewer #3: I am happy and glad to the editor for sending me the paper for reviewing.

I am happy reading the version where authors addressed an important topic in the context of India. India is a large country with geographical heterogeneity. Author clearly showed some geographical differences in terms of undernutrition of children. Although the dataset is very old I think the paper carrying important guidelines for further investigation.

My short specific comments are following.

What is the power of the analysis of the paper? Why didn’t you do multilevel analysis as it has cluster effects?

Now 2021 but you used almost ten years oldest dataset over the passing of times is the prevalence is same or have been changed?

The conclusion of the abstract is written in more general fashion please be specific relying on your findings.

Authors didn’t mention any limitations of the study!

What is the main rationality of the study that is missing in the background section.

6. PLOS authors have the option to publish the peer review history of their article (what does this mean?). If published, this will include your full peer review and any attached files.

Reviewer #1: No

Reviewer #2: No

Reviewer #3: No

---

## [Author Response · Author response to Decision Letter 0]

21 Apr 2022

Dear Academic Editor

PLOS ONE

Thank you for giving us an opportunity to strengthen the paper by addressing suggestions from you and reviewers. We have made the necessary changes to the manuscript. Following is our response to the comments. 

Editor’s comments:

1. Elaboration of a paragraph with the limitations of the study.

Response: Thank you for this suggestion. The paragraph on limitation has been elaborated (lines 548 to 557)

2. Discussion about possible differences between the period of data collection and the present day.

Response: Thank you for the comment. The CCM-II data conducted in 2014-15 at the level of sub-districts, was the first and latest of its kind yet. And the unit-level district level NFHS-4 data is also the latest dataset available for now. The unit-level NFHS-5 dataset for the year 2018-2020 has not been released yet. The aggregate level Odisha factsheet of this 5th wave is released and the changes in the prevalence over the years have been mentioned in the main text- lines 141 to 144.

3. Explain that the spatial lag model and Spatial Error Model are the CAR and SAR models.

Response: Thank you for your suggestion. The SEM model explanation was mentioned in the earlier version, the explanation of Spatial Lag model is now included in the revised version (lines 251 to 273). The spatial autoregressive models are the regressions models which account for spatial autocorrelation in the dataset. To the best of our review, literatures commonly refer to three spatial autoregressive models, which are: the conditional autoregressive model (CAR), the simultaneous autoregressive model (SAR), and the moving average (MA) process model. Two special cases of the simultaneous autoregressive (SAR) models, include the spatial lag model (SLM), and spatial error model (SEM), which were utilised in the current study to model the variations. The lines 372 to 375 also explains how SEM proved to be an improved model than the OLS and SLM, based on the model diagnostics.

4. Inform the distribution of the data in the spatial regressions. Check if other distributions (eg Poisson or negative binomial) have a better fit.

Response: Thank you for the suggestion. The distribution of the data was normal, which is also one of the crucial assumptions in the OLS model. This has been mentioned in the revised version. In the SAR process, the correlated error term is assumed to have a normally distributed η. The Jarque–Bera test, a goodness-of-fit test to check whether sample data have the skewness and kurtosis matching a normal distribution, was applied. The JB test value of 4.34 with 0.114 p-value, proved that the distribution was normal. The Shapiro-Wilk W test for normal data also resulted in having probability of 0.081, thus, indicating the normality of the data.

5. Implement or justify the non-implementation of multilevel models.

Response: We thank the reviewer for this insightful comment. Multilevel modelling was out of the scope of the current study. The vision of the present study was to explore, identify and examine the clusters of high and low burden of malnutrition with its correlates, across the blocks of Odisha. Using the geo-spatial techniques, we were able to identify the clusters and outliers of the parameters, thus helping us to identify the high-risk areas. And based on these LISA results, we could observe how the three dominant clusters coincided with the three natural regions of the state as classified by the National Sample Survey organisation based on similar agro-climatic conditions, cultural attributes, population densities, etc. Then further, multivariate analyses were conducted to study the different socio-economic, demographic, maternal and child related factors impacting the nutritional status of the child, specific to these three geographical regions of the state; i.e, to study how the association and impact varied across and within the three regions. This approach may not have been fulfilled using multilevel models, since that would have only helped us study the variability across the levels in terms of region, households and individual indicators, but not specific to the three different geographical regions. However, in our future work we would try to explore multilevel modelling complemented with other appropriate methods to see how the findings vary, if they vary at all.

Journal Requirements:

Response: We have reviewed the journal guidelines and have made corrections wherever it was required. 

Response: This study compiled and extracted data information from unit-level data from the NFHS (2015–16) for India, which is publicly available and with no access to personal identifiers. Hence, no need of ethical approval for conducting this study. The CCM-II (2014-15) aggregate level data for Odisha, utilised in the study, is publicly available in their technical report [Odisha Technical and Management Support Team. (2015). Concurrent Monitoring II: Odisha state survey 2014-15. Options Consultancy Services LTD, IPE Global, CARE India. Available from: https://options.co.uk/sites/default/files/ccm_ii_final_-_2.pdf (accessed on 12 April 2021)], however, we have also received the approval in principle by W&CD & Mission Shakti Department, Government of Odisha. This survey received ethical approval from the Indian Institute of Public Health, as mentioned in their technical report. Hence, no need of ethical approval for conducting this study. As required, the ethics statement has been mentioned in the ‘Data and Methods’ section of the revised manuscript.

3. We note that Figures 1-5 and S1 file in your submission contain map/satellite images which may be copyrighted. All PLOS content is published under the Creative Commons Attribution License (CC BY 4.0), which means that the manuscript, images, and Supporting Information files will be freely available online, and any third party is permitted to access, download, copy, distribute, and use these materials in any way, even commercially, with proper attribution. For these reasons, we cannot publish previously copyrighted maps or satellite images created using proprietary data, such as Google software (Google Maps, Street View, and Earth). For more information, see our copyright guidelines: http://journals.plos.org/plosone/s/licenses-and-copyright.

a) You may seek permission from the original copyright holder of Figures 1-5 and S1 file to publish the content specifically under the CC BY 4.0 license. 

Response: The figures are authors’ own work; hence, we don’t need to take any permission. These maps were created using shape files which were made available by the Spatial Planning and Analysis Research Centre Pvt. Ltd, Odisha. The block-wise shapefiles could also be found publicly at “Ministry of Rural Development, 2022. PMGSY Rural Connectivity Datasets, https://geosadak-pmgsy.nic.in/opendata/. Published under India’s Government Open Data License: https://data.gov.in/government-open-data-license-india ” which was last accessed on 27th February 2022.

Reviewers' comments:

I. Reviewer #1: The article presents important data on child malnutrition in India, with geospatial analysis associated with socioeconomic and health characteristics of the population according to region of residence. Its results are of relevance to the area of Nutrition and Public Health. The article is properly written in formal English and has a clear and direct format.

Response: Many thanks for your kind review and supportive comments.

II. Reviewer #2: Objective: The aim of the study was to investigate undernutrition among children in Odisha, Indian, at sub-district levels using data from the Concurrent Child Monitoring Survey-II, 2014-15, through a geospatial analysis regional, state, district, urban-rural.

Comments:

This is an important study, of an unprecedented nature in the country, in which this analysis of the spatial distribution of child malnutrition by block or sub-district was carried out for the first time. Therefore, it is aligned with the scope of the journal Plos One.

- The methods are adequately described, allowing the reproduction of the study and providing a clear display for the results.

- The results are properly presented and the discussion dialogues with the scientific literature

- Conclusion: the authors highlight the main findings and contributions of the study to adress public policies to combat child malnutrition in different regions India. Below are some considerations.

- Considering the persistense of undernutrition as a public heath problem in Índia, and the originality of this study, the manuscript could be eligible for publication in Plos One, after a few changes listed below:

Response: We are thankful for all your supportive and constructive comments, these have helped us to strengthen the manuscript. Below are the responses to each of your comments:

1. Abstract: In order to facilitate the communication with the scientific readers, I suggest the authors to build a structured Abstract: Background; Objective; Methods; Results; Conclusion.

Response: Thank you for suggesting to write the abstract in the said structured way. As suggested, we have now structured the abstract into Background; Data and Objective; Methods; Results; Conclusion.

2. Introduction: I suggest shifting the items: “Data and Methods, Data, Variable Description” to the first paragraph of the methods section. Building a flowchart could help improving the conciseness of the Methods.

Response: Many thanks for your valuable suggestion. The current flow of the data and methods is as per the journal structure. As suggested, a flowchart of the methods have been built and added as a supplementary file.

3. Figures and Tables: It is necessary to supply complete titles and legends for all figures and Tables, including the country and the year of the study.

Response: Thanks for the comment. The complete titles having state, country and year of the study have been changed, as suggested. The legends are included along with the figures.

III. Reviewer #3: I am happy and glad to the editor for sending me the paper for reviewing.

I am happy reading the version where authors addressed an important topic in the context of India. India is a large country with geographical heterogeneity. Author clearly showed some geographical differences in terms of undernutrition of children. Although the dataset is very old I think the paper carrying important guidelines for further investigation.

My short specific comments are following.

Response: We are thankful for all your supportive and constructive comments, these have helped us to strengthen the manuscript. Below are the responses to each of your comments:

1. What is the power of the analysis of the paper? Why didn’t you do multilevel analysis as it has cluster effects?

Response: Many thanks for your comments. The study tried to investigate the statistically significant clusters and spatial patterns of high and low burden of undernutrition among children and its correlates at the level of block or sub-districts using the Concurrent Child Monitoring Survey-II, 2014-15, which is the first of its kind. Further, the second part of the analysis is the region-specific study, where the impact of various demographic, socio-economic and maternal factors on the prevalence of undernutrition were studied, specific to the three broad geographical regions of the state; and to the best of our review, this methodology of a micro-level, region-specific analytic approach, for evidence-based planning, has been conducted for the first time in the Indian context.

Additionally, a few of the highlights of the study includes: (1) the spatial variation and their occurrence in specific clusters that happen to be contiguous homogenous regions with respect to geographical features, rural population densities, crop-pattern, etc. (2) The parameters requiring attention from a programme implementation perspective differ across the development ladder. (3) The findings carry important signalling effects for local and regional decision-makers and necessitate the need to develop a region-specific evidence-base at the micro level to develop a customised plan of action for each area.

Multilevel modelling was out of the scope of the current study. The vision of the present study was to explore, identify and examine the clusters of high and low burden of malnutrition with its correlates, across the blocks of Odisha. Using the geo-spatial techniques, we were able to identify the clusters and outliers of the parameters, thus helping us to identify the high-risk areas. And based on these LISA results, we could observe how the three dominant clusters coincided with the three natural regions of the state as classified by the National Sample Survey organisation based on similar agro-climatic conditions, cultural attributes, population densities, etc. Then further, multivariate analyses were conducted to study the different socio-economic, demographic, maternal and child related factors impacting the nutritional status of the child, specific to these three geographical regions of the state; i.e, to study how the association and impact varied across and within the three regions. This approach may not have been fulfilled using multilevel models, since that would have only helped us study the variability across the levels in terms of region, households and individual indicators, but not specific to the three different geographical regions. However, in our future work we would try to explore multilevel modelling complemented with other appropriate methods to see how the findings vary, if they vary at all.

2. Now 2021 but you used almost ten years oldest dataset over the passing of times is the prevalence is same or have been changed?

Response: Thank you for the comment. We agree, the dataset is ten years old but unfortunately, no data on the nutritional status of the children at the level of sub-districts are available post this period. That is, the CCM-II data conducted in 2014-15 at the level of sub-districts, was the last of its kind yet. And the unit-level district level NFHS-4 data is also the latest dataset available as of now. The unit-level NFHS-5 dataset for the year 2018-2020 has not been released yet, only the aggregate level Odisha factsheet of this 5th wave is released. Also, even though the fourth round of NFHS (and now the 5th) provide unit level data, these are not representative at the block level, which CCM-II is. Hence, the unit-level data of the recent NFHS, cannot be used for a sub-district level analysis.

The changes in the prevalence over the years have now been mentioned in the main text- line 141 to 144.

Also, in India, good data set at sub-district level is not available from any other survey so far, and even if it is 10 years old, the analysis may persuade policy makers in individual states and perhaps in Odisha as well to carry out similar surveys so as to adopt an informed regional strategy to tackle child malnutrition.

3. The conclusion of the abstract is written in more general fashion please be specific relying on your findings.

Response: Thanks for the suggestion. As advised, the conclusion of the abstract has been modified accordingly.

4. Authors didn’t mention any limitations of the study!

Response: Thank you for highlighting this aspect. The paragraph of limitation has been elaborated now (lines 548 to 557)

5. What is the main rationality of the study that is missing in the background section.

Response: Thank you for this comment. The availability of data and its analysis at the micro-level pose a challenge. And, given the regional variation in the nutritional status of children, there is a strong need for integration of GIS tools with micro level data to identify pockets of low and high levels of malnutrition to get a more nuanced understanding of the problem. This will enable customized and evidence-based planning to produce malnutrition free enclaves with consequent ripple effect in space and in time. Hence, studies based on micro level data is required which will help policy makers in developing localized policies and programs. Through our comprehensive and critical literature review, we found (to the best of our search) that none of the studies have provided spatial insights at the sub-district level, mainly because of the lack of data availability. The present study aimed to bridge this gap by conducting spatial analysis for a state at a micro level-- using the block/ sub-district as the spatial unit and further identified the correlates of the children undernourished specific to the clusters of high-low burden. And to the best of our review, this methodology of a micro-level, region-specific analytic approach, for evidence-based planning, has been conducted for the first time in the Indian context. These points have been incorporated in the background section of the manuscript.

---

## [Decision Letter · Decision Letter 1]

3 May 2022

Region matters: Mapping the contours of undernourishment among children in Odisha, India

PONE-D-22-02352R1

Dear Dr. Agnihotri,

We’re pleased to inform you that your manuscript has been judged scientifically suitable for publication and will be formally accepted for publication once it meets all outstanding technical requirements.

Kind regards,

Vinícius Silva Belo

Academic Editor

PLOS ONE

Reviewers' comments:

Reviewer's Responses to Questions

**Comments to the Author**

1. If the authors have adequately addressed your comments raised in a previous round of review and you feel that this manuscript is now acceptable for publication, you may indicate that here to bypass the “Comments to the Author” section, enter your conflict of interest statement in the “Confidential to Editor” section, and submit your "Accept" recommendation.

Reviewer #3: All comments have been addressed

2. Is the manuscript technically sound, and do the data support the conclusions?

Reviewer #3: Yes

3. Has the statistical analysis been performed appropriately and rigorously? 

Reviewer #3: Yes

4. Have the authors made all data underlying the findings in their manuscript fully available?

Reviewer #3: Yes

5. Is the manuscript presented in an intelligible fashion and written in standard English?

Reviewer #3: Yes

6. Review Comments to the Author

Reviewer #3: Thanks to you all for addressing the comments I made previously. This article is important for further taking any initiative to reduce malnutrition in childhood.

7. PLOS authors have the option to publish the peer review history of their article (what does this mean?). If published, this will include your full peer review and any attached files.

Reviewer #3: No

---

## [Editor Report · Acceptance letter]

10 May 2022

PONE-D-22-02352R1 

Region matters: Mapping the contours of undernourishment among children in Odisha, India 

Dear Dr. Agnihotri:

I'm pleased to inform you that your manuscript has been deemed suitable for publication in PLOS ONE. Congratulations! Your manuscript is now with our production department. 

Kind regards, 

on behalf of

Dr. Vinícius Silva Belo 

Academic Editor

PLOS ONE